# Is There a Bidirectional Association between Polycystic Ovarian Syndrome and Periodontitis? A Systematic Review and Meta-analysis

**DOI:** 10.3390/jcm9061961

**Published:** 2020-06-23

**Authors:** Vanessa Machado, Cláudia Escalda, Luís Proença, José João Mendes, João Botelho

**Affiliations:** 1Clinical Research Unit (CRU), Centro de Investigação Interdisciplinar Egas Moniz (CiiEM), Instituto Universitário Egas Moniz (IUEM), 2829-511 Caparica, Portugal; claudia.escalda4@gmail.com (C.E.); jmendes@egasmoniz.edu.pt (J.J.M.); jbotelho@egasmoniz.edu.pt (J.B.); 2Quantitative Methods for Health Research (MQIS), Centro de Investigação Interdisciplinar Egas Moniz (CiiEM), Instituto Universitário Egas Moniz (IUEM), 2829-511 Caparica, Portugal; lproenca@egasmoniz.edu.pt

**Keywords:** polycystic ovary syndrome, periodontitis, periodontal disease, systematic review, meta-analysis

## Abstract

Polycystic ovary syndrome (PCOS) has reproductive and metabolic properties that may be linked to periodontitis (PD). This study aimed to update and render a robust critical assessment on all evidence linking PCOS and PD, and appraising a hypothetical bidirectional association. Five databases (PubMed, Scholar, EMBASE, Web of Science and CENTRAL) were searched up to May 2020. Case-control and cohort studies on the association of PCOS and PD were included. The risk of bias of observational studies was assessed through the Newcastle-Ottawa Scale (NOS). Random effects meta-analyses of standardized mean difference (SMD) and risk ratio (RR) were performed. We followed Strength of Recommendation Taxonomy (SORT) to appraise the strength and quality of the evidence. Twelve case-controls fulfilled the inclusion criteria (876 with PCOS and 48170 healthy controls), all scored as having a low risk of bias. Meta-analysis revealed that PCOS females have 28% more risk towards PD, and PD females have 46% more risk to have PCOS. PCOS females with PD had higher gum bleeding, periodontal pocket depth and clinical attachment loss than non-PCOS females with PD. Populations with undefined periodontal status contribute to underestimated results. On the basis of the available evidence, it is possible to assume a bidirectional link between PCOS and PD. That is, PCOS increases by 28% the risk of having PD and in the same fashion, PD increases by 46% the risk of having PCOS. Furthermore, women with PCOS were associated with worsening clinical characteristics and inflammation of PD. These findings suggest that PCOS and PD may be linked. Hence, further prospective and clinical trial studies with nonsurgical periodontal therapy are necessary to clarify the existence of an increased risk of PCOS in women with PD and vice-versa.

## 1. Introduction

Polycystic ovary syndrome (PCOS) is a complex endocrine, reproductive and metabolic condition, with a worldwide prevalence ranging 5–15% [1]. PCOS is characterized by polycystic ovaries, hyperandrogenism with impaired gonadotropin secretory activity, hyperinsulinemia, hypothalamic–pituitary–ovarian (HPO) axis changes, and ovulatory and menstrual dysfunction [2,3,4,5,6,7,8]. Additionally, PCOS is also accompanied by psychological alterations, such as anxiety, depression and poor quality of life [9]. The pathogenesis of PCOS remains to be fully understood due to its multifactorial profile. Nevertheless, chronic infections have gained relevance, for instance, because of the associated high levels of oxidative stress (OS), inflammatory molecules, adhesion molecules, as well as blood lymphocytes and monocytes. In particular, the proinflammatory state has been one of the most investigated for the PCOS link with periodontitis.

Periodontitis (PD) is a chronic multifactorial inflammatory condition caused by a dysbiotic plaque and causes the destruction of the periodontium, the tissues that surround teeth [10,11]. Periodontitis is a pandemic noncommunicable disease and its symptoms start with gum inflammation (gingivitis) to an exacerbated and uncontrolled inflammatory response from the innate and adaptive immune system [12,13]. The destruction of the periodontium allows for bacteria and bacterial products to gain access to the systemic circulation through the ulcerated epithelium and destruction of the periodontium.

Over the last decade, cross-sectional evidence has suggested that PCOS patients may be more prone to develop periodontitis [14,15,16,17,18,19,20,21,22] and with higher local levels of OS [14,17]. Also, a randomized clinical trial showed that nonsurgical periodontal therapy together with myo-inositol therapy contributed to a higher decrease of systemic inflammatory levels compared with myo-inositol solely in women with PCOS and periodontitis [23]. Furthermore, three systematic reviews have supported a possible association between PCOS and PD, however none of them were able to synthesize estimates on this association nor the impact on clinical characteristics [9,24,25]. However, there is inconclusive evidence regarding the risk of PCOS patients to develop periodontitis and vice-versa, in a potential bidirectional relationship, and the potential role of PCOS associated inflammation on periodontal tissue health (worsen or better the gingival condition).

Given the recent increase of newly available studies, the primary aim of the present review was to update and render a robust critical assessment on all evidence linking PCOS and periodontitis, and appraising a hypothetical bidirectional association, following a meticulous systematic protocol.

## 2. Materials and Methods

### 2.1. Protocol

This systematic review was defined *a priori* and was performed following the Cochrane Handbook of Systematic Reviews of Interventions [25] and reported according to the Preferred Reporting Items for Systematic Reviews and Meta-Analyses (PRISMA) guidelines [26].

### 2.2. Focused Question and Eligibility Criteria

We set the following broad research question so as not to introduce bias between PCOS and Periodontitis. The following focused question was addressed: “Is there a bidirectional association between PCOS and Periodontitis?”, with the following specific questions: “Does PCOS have an effect on the healthy periodontium?”“Does PCOS influence periodontal clinical characteristics of Periodontitis?“Does Periodontitis influence clinical characteristics of PCOS?”

We developed a protocol to answer the following PECO question:Female patients without PD (Patients-P); PCOS (Exposure-I); No PCOS (Comparison-C); periodontal probing depth (PPD), clinical attachment loss (CAL), bleeding on probing (BoP) levels (Outcome-O)Female patients with PD (Patients-P); PCOS (Exposure-I); No PCOS (Comparison-C); PPD, CAL, BoP levels (Outcome-O)Female patients with PD (Patients-P); PCOS (Exposure-I); No PCOS (Comparison-C); endocrine outcomes (Ferriman–Gallwey score, free androgen index, dehydroepiandrosterone sulfate (DHEAS), free testosterone and total testosterone levels); glycemic (fasting blood insulin, fasting blood glucose, homeostatic model assessment (HOMA-IR)); and physical (waist-to-hip ratio (WHR)) outcomes - O

To address the first PECO research question, studies reporting on the presence of PPD, CAL and BoP evaluation in PCOS female patients and healthy controls with non-PD diagnosis, were included. Studies comparing these two groups of patients without clinical and radiophonic diagnostic confirmation in the control group were excluded.

For the second and third PECO questions, studies reporting the presence of PPD, CAL and BoP values of PCOS patients and healthy controls were included. Studies that have included PCOS patients and healthy controls without previously knowing their periodontal status following periodontal clinical assessment were excluded due to the high risk of reporting bias.

### 2.3. Search Strategy

We searched in Pubmed, Medical Literature Analysis and Retrieval System Online (MEDLINE), CENTRAL (The Cochrane Central Register of Controlled Trials), EMBASE, Web of Science from the earliest data available until March 2020, and last updated in May 2020. We merged keywords and subject headings in accordance with the thesaurus of each database and applied exploded subject headings. Our Pubmed search strategy was based on the following algorithm: (chronic periodontitis OR periodontitis, chronic OR adult periodontitis OR periodontitis, adult OR periodontal disease OR alveolar bone loss OR attachment loss, periodontal OR periodontal pocket) and (polycystic ovary syndrome or PCOS). Gray literature was searched through the OpenGrey portal [27]. Additional relevant literature was included after a manual search of the reference lists of the final included articles. If there were multiple publications for the same study, data from the largest sample were used. The study type was restricted to human studies. We included both randomized clinical trials (RCTs) and non-RCTs (case-control, cohort studies and cross-sectional) that reported on (A) PCOS female patients with and without associated PD and/or (B) patients with PD with and without a concomitant PCOS diagnosis. The authors were contacted when necessary additional data clarification was required.

### 2.4. Study Processs

Two independent investigators (V.M. and J.B.) screened the titles and abstracts of the retrieved studies. We resolved disagreement through adjudication or by a third reviewer (J.J.M.). The final selection of studies was independently performed by three authors who reviewed the full text of the selected papers based on the inclusion criteria mentioned above. Any disagreements were resolved by discussion.

A predefined table was used to extract necessary data from each eligible study, including the first author’s name, study design, publication year, the country of origin), inclusion/exclusion criteria, number of female participants, mean age in year, periodontal case definition, and PCOS diagnosis based on validated classification criteria. Clinical periodontal measures included: PPD, CAL, BoP, missing teeth. All data were independently extracted by two reviewers with a consensus in all aspects. Concerning additional data clarifications, we attempted to contact the corresponding authors twice, with an interval time of one week, without success.

### 2.5. Risk of Bias (RoB) in Individual Studies

Methodological quality was appraised using the Newcastle-Ottawa (NOS) Scale by one calibrated author. In this last tool, case-control and cohort studies are scored across three categories: studies with 7–9 stars of low RoB, studies with 5–6 stars of moderate RoB, whilst studies with less than 5 stars were deemed as being of high RoB. If any doubt occurred, they were resolved by discussion with a second author.

### 2.6. Statistical Analysis

All statistical analyses were performed in R version 3.4.1 (R Studio Team 2018) using a DerSimonian-Laird random-effects model [28]. All random-effects meta-analysis and forest plots were performed using ‘meta’ package [29,30]. Risk ratio (RR), 95% confidence intervals (CI) and the weight for each included study were calculated for the risk of PCOS towards PD and vice-versa. For continuous data, mean values and standard deviations (SD) were used and analyzed with standard mean differences (SMD) and 95% CI. We used mm as the unit of measurement in the SMD meta-analysis. In the case of median and interquartile range reported, we converted to mean and SD following Hozo procedure [31]. The data was reported regardless of the individual case definition. Subgroup meta-analysis estimates were pooled according to samples with PD, without PD and with undefined periodontal status (that is, studies where periodontal diagnosis was not performed, but measures of PPD and CAL were recorded). Statistical heterogeneity was inspected through I^2^ index and Cochrane’s Q statistic (p < 0.1), and the overall homogeneity was calculated through the χ2 test [25]. All tests were two-tailed, with alpha set at 0.05. Further, the weight percentage given to each study in each analysis was provided in the forest plots. Publication bias was planned if, at least, 10 or more studies were included [25]. The overall estimates were reported with 95% CI. 

### 2.7. Strength of Recommendation

We used the SORT (Strength of Recommendation Taxonomy) to judge the strength and quality of the evidence [32]. We discussed the outcomes of the present systematic review, clinical recommendations, and future necessary research.

## 3. Results

### 3.1. Characteristics of Included Studies

The search strategy identified a total of 239 possibly relevant articles. After duplicates removal, 54 papers were judged against the eligibility criteria, and 185 were excluded after titles and/or abstracts review. Out of these 19 articles which were subjected to full paper review eligibility, 7 articles were excluded as they did not address the research questions (Appendix A). As a result, a final number of 12 case-control studies [14,15,16,17,18,19,21,22,33,34,35,36] met all of the inclusion and exclusion criteria and were included for qualitative synthesis (Figure 1). Interexaminer reliability at the full-text screening was considered very substantial (kappa score = 1.00).

The characteristics of the included studies are shown in Table 1. We identified 12 case-control studies from five different countries, across Europe and Asia. These studies were published between 2011 and 2020. The sample sizes ranged from 40 [35] to 48.820 participants [18] per study. A total of 49.965 participants were included in this review, comprising 768 PCOS female individuals without PD, 613 PCOS female individuals with PD, 24.152 female patients with PD and 24.445 healthy controls.

### 3.2. Methodological Quality

The RoB for observational studies, with Newcastle-Ottawa scale, varied across the studies, ranging from 7 to 9 stars (Appendix A). After appraisal, four articles presented the highest score (9/9) [15,19,21,22]. Furthermore, six articles scored 8/9 [16,17,33,34,35,36]. Almost half of the articles failed in including a representative amount of cases (41.7%, n = 5), nevertheless, most of them made a good definition of the cases (91.7%, n = 11) as well as a good selection (91.7%, n = 11) and definition of the controls (75.0%, n = 9). All studies had excellent comparability results, secure record of ascertainment of exposure, equal ascertainment method for cases and controls and the same nonresponse rate for both cases and controls (100.0%, n = 12).

### 3.3. Methodological Quality

#### 3.3.1. Bidirectional Association between Polycystic Ovarian Syndrome and Periodontitis

To investigate the likelihood of PCOS female individuals to have PD, three studies were included [15,18,21], in a total of 49,046 participants. The overall results reveal that PCOS individuals have on average 28% more risk to develop periodontitis (RR [95% CI]: 1.28 [1.06–1.55], p < 0.0001, I^2^ = 43.0%) with moderate heterogeneity, revealing some variation among the included studies (Figure 2).

To investigate the likelihood of female individuals with periodontitis to have PCOS, the same three studies were included [15,18,21]. The overall results reveal that females with periodontitis have, on average, 46% more risk to be diagnosed with PCOS (RR [95% CI]: 1.46 [1.29–1.66], p < 0.0001, I^2^ = 0.0%), with complete homogeneity among the included studies (Figure 3).

#### 3.3.2. PCOS Effect on the Gingival Inflammation

For the BoP analysis, ten studies were included to investigate the association of PCOS on gingival bleeding [14,15,16,17,19,21,22,33,34,36]. Five studies compared BoP levels of PD patients with and without PCOS [16,17,21,33,34], four studies compared BoP levels of non-PD patients with and without PCOS [17,21,34,36] and four studies compared BoP between PCOS and controls [14,15,19,22] (Figure 4). Overall, this analysis had 594 and 490 individuals with PCOS and without PCOS, respectively. A significant MD was found (MD [95% CI]: 2.21 [1.32–3.11] p < 0.01), and the heterogeneity among studies was high (98% and 94%, respectively). Specifically, PD patients with PCOS presented more gingival inflammation that PD females without PCOS (MD [95% CI]: 1.56 [1.07–2.05] p = 0.75), with complete homogeneity (I^2^ = 0.0%).

#### 3.3.3. PCOS effect on Periodontal Structure Loss

For the assessment of PPD, nine studies were included to investigate the association of PCOS on PPD [14,15,16,17,19,21,22,33,34,36]. Four studies compared PPD levels of PD patients with and without PCOS [17,21,33,34], four studies compared PPD levels of non-PD patients with and without PCOS [17,21,35,36] and three studies compared PPD between PCOS and controls [14,15,22] (Figure 5). Overall, this analysis had 507 and 354 individuals with PCOS and without PCOS, respectively. A significant MD was found (MD [95% CI]: 0.61 [0.28–0.94] p < 0.01), and the heterogeneity among studies was high (I^2^ = 96%). Subgroup analysis was performed, and women with PCOS had greater PPD levels than women without PCOS (MD [95% CI]: 0.98 [0.15–1.81] p < 0.01).

For the assessment of CAL, eight studies were included to investigate the association of PCOS on CAL [14,15,16,17,21,22,33,34]. Five studies compared CAL levels of PD patients with and without PCOS [16,17,21,33,34], two studies compared CAL levels of non-PD patients with and without PCOS [17,21] and three studies compared CAL between PCOS and controls [14,15,22] (Figure 5). Overall, this analysis had 464 and 395 individuals with PCOS and without PCOS, respectively. A significant MD was found (MD [95% CI]: 0.33 [0.13–0.53] p < 0.01), and the heterogeneity among studies was high (I^2^ = 92%). Subgroup analysis was performed and women with PCOS had greater CAL levels than women without PCOS (MD [95% CI]: 0.51 [0.12–0.89] p < 0.01) (Figure 6).

#### 3.3.4. Additional Analyses

Considering the minimum number of 10 studies, publication bias analyses was not possible.

According to the SORT recommendation, the evidence from observational studies revealed that PCOS and PD are strongly associated, based on consistent findings of at least two good-quality meta-analyses that obtained significant results and complete homogeneity (SORT A) [32].

## 4. Discussion

### 4.1. Summary of Main Findings and Quality of The Evidence

This systematic review provides an association of PCOS and PD, with an overall SORT A recommendation. Despite the number of reports, our systematic review is the first to investigate the likelihood of a bidirectional link between PCOS and PD. These results emphasize the importance of the association between PD and the clinical status of PCOS women as a common shared low-grade systemic inflammation status.

Overall, the results of this systematic review support a bidirectional link between PCOS and PD. That is, PCOS increases by 28% the risk of having PD and in the same fashion, PD increases by 46% the risk of having PCOS. It is important to emphasize that these results had very interesting degrees of heterogeneity, however, they need to be carefully interpreted due to the low number of articles available. This bidirectional association demands further studies because we were only able to infer how the presence of PCOS links with some periodontal characteristics (such as gingival inflammation and periodontal structure loss), rather than ascribe how the variation of the PD condition influences PCOS clinical characteristics.

Further, PCOS females with PD had higher gingival inflammation and periodontal structure loss than non-PCOS females with PD. While the results are in line with a previous systematic review [25], our outcomes regarding BoP on PD females (PCOS vs. non-PCOS) present complete homogeneity. Also, for the remaining measures (PPD and CAL), our results showed higher heterogeneity levels, which might be explained by the increased the number of studies in each meta-analysis.

Comprehensively, the link between PD and PCOS is based on the chronic subclinical inflamed status of both conditions [18,37]. In other words, a persistent subclinical inflammation triggers the synthesis of a panel of proinflammatory markers (such as C-Reactive Protein (CRP), tumor necrosis factor-α, interleukin (IL)-6, IL-17, and matrix metalloproteinase-9) [36,38,39], and potentiates an oxidative stress environment (through local oxidant status-like myeloperoxidase and nitric oxide) [14,17,36]. Unfortunately, this systematic review was not able to provide pooled estimates for these inflammatory markers within the PCOS–PD axis due to the lack of studies. In this sense, further investigations on inflammation are of the utmost importance because they may explain the biological mechanisms connecting these two entities. Additionally, PD management has been implicated in insulin levels control with periodontal treatment allowing the alleviation of high glycemic levels [40], and therefore uncontrolled periodontitis may indirectly impact on PCOS clinical status.

### 4.2. Strengths and Potential Limitations

This systematic review followed a rigorous protocol, with up-to-date international reporting guidelines, a thorough literature search and all included articles were of low risk of bias. There are, however, a number of limitations to mention. The results are derived from observational studies denting the inference of causality. In addition, the case definition of PD among the included studies was poorly uniform and may explain the elevated heterogeneity levels. Even so, the diagnosis of PCOS was mostly based on the Rotterdam criteria 2003 [41], which may have minimized the degree of heterogeneity, except for the study Tong et al. [18] that used the ICD-9-CM system. Another limitation is that the included articles are not coherent on the blood analysis and physical assessments of PCOS so we can only have an objective insight of the periodontal parameters, plus, the articles vary considerably on the number of subjects included, which may be a cause of heterogeneity. It is also important to mention that some articles did not exclude patients with gingivitis from the periodontally healthy groups which may lead to differences in the results.

Importantly, a novelty of our study is that our approach grouped whether studies considered the periodontal status of the included patients, differently from the systematic review by Farook et al. [25]. In other words, we intended to differentiate studies that had made periodontal diagnosis a priori from those that included PCOS and control patients regardless of the periodontal status. Our results confirm that the pooled estimates from studies with undefined periodontal status are more likely to produce lower differences and more heterogeneous results. Interestingly, no study used the most recent classification on PD [42] and is a limiting factor since it is highly advisable that studies use updated classifications. Furthermore, the new PD case definition presents a potential improved ability to transmit the entire periodontal condition [43,44]. Therefore, future studies on PCOS and PD should diagnose the periodontal status of the included participants and follow the later PD diagnosis definition.

### 4.3. Interpretation and Clinical Implications

The cause of PCOS remains unknown [7], however is characterized as a syndrome related to the interplay of genetic and environmental factors [45]. From this standpoint, metabolic abnormalities associated with insulin resistance have become increasingly relevant [45,46,47]. Metabolic syndrome (MetS) is defined by a clump of hyperglycemia, obesity, dyslipidemia, and hypertension [45], and several clinical studies have reported marked prevalence among PCOS patients [48,49,50,51,52,53,54].

Compensatory hyperinsulinemia caused by MetS nettles excessive action of insulin, which is a major extraovarian factor in the steroidogenic dysregulation in PCOS [53,55,56,57]. Comprehensively, excessive insulin overwhelms theca cell steroidogenesis [58,59,60], inducing abnormal production of ovarian androgens and causes subsequent ovarian dysfunction [45,61,62]. PD and MetS have been consistently associated in the past two decades [63,64]. Furthermore, the relationship between PD and poor glycemic control is well-established in a bidirectional fashion based on the impact of chronic low-grade systemic inflammation [10,65,66,67,68,69,70,71,72,73,74,75,76,77,78,79]. Similar to MetS, this long standing inflamed status is closely involved with abnormal hyperglycemia levels, poor glycemic control, and insulin resistance [68,80]. Our results support this hypothesis, showing a bidirectional association between PCOS and PD, however we were not able to provide further analysis to establish mechanistic relationships between these conditions, and therefore more research is warranted to clarify this link.

The complexity of cytokine networks and oxidative stress in periodontal pathogenesis is becoming increasingly evident. In addition, insulin resistance is central in the development of PCOS due to stimulation of androgens in theca cells and consequently remaining a mechanism of PCOS. The sum of all periodontal factors in women with PCOS can lead to exacerbation of its clinical characteristics, (hyperandrogenism, ovulatory dysfunction, and polycystic ovarian morphologic features). Further, PD treatment can be a coadjuvant in PCOS therapy, since one randomized clinical trial showed that periodontal treatment together with a myo-inositol regimen greatly reduced the systemic inflammatory burden [23]. In other words, periodontal care may aid the clinical management of PCOS and that constitutes a paradigm shift in the multidisciplinary approach of this condition.

### 4.4. Research Implications

This systematic review provides evidence for a possible bidirectional association between PCOS and PD. These results shed light on the need for more studies further investigating the underlying mechanisms between these two conditions. Future investigations should incorporate serum and physical analysis of the main biomarkers of PCOS (namely for hyperandrogenemia, testosterone and insulin resistance) in their design. Also, studies should divide participants according to the PCOS and PD status in longer prospective study designs, to ensure more evidence on the impact between these diseases.

In addition, new clinical trials assessing the impact of nonsurgical periodontal treatment (NSPT) on PCOS patients are encouraged to clarify if treating PD might alleviate PCOS clinical course (biomarkers for reproductive and metabolic health) and impact its risk factors (e.g., insulin resistance).

The development and implementation of a core set of research guidelines on PCOS and PD are highly recommended to ensure that results are collected and reported in a similar and consistent manner to allow for future definite conclusions on this association.

## 5. Conclusions

Within the limitations of this review, our results show that there is a higher risk of individuals with PCOS being diagnosed with PD and vice-versa. PCOS may worsen the periodontal condition as higher levels of gingival inflammation, PPD and CAL were found. The opposite cannot be said as the PCOS variables are not fully elucidative. Obstetrician-gynecologists and dentists should collaborate and improve the clinical outcomes of their PCOS patients.

## Figures and Tables

**Figure 1 jcm-09-01961-f001:**
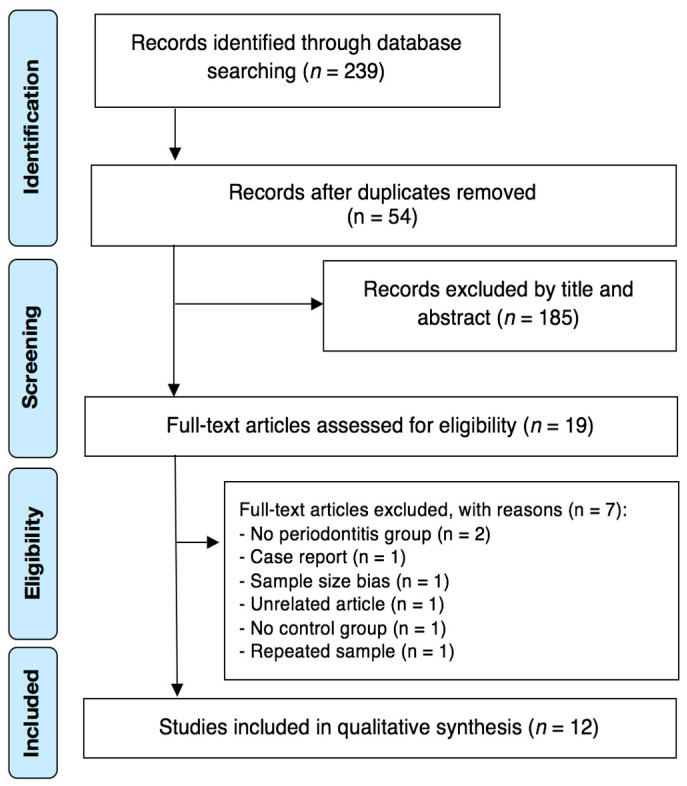
Article selection flow chart for the systematic review according to Preferred Reporting Items for Systematic Reviews and Meta-Analyses (PRISMA) guidelines.

**Figure 2 jcm-09-01961-f002:**
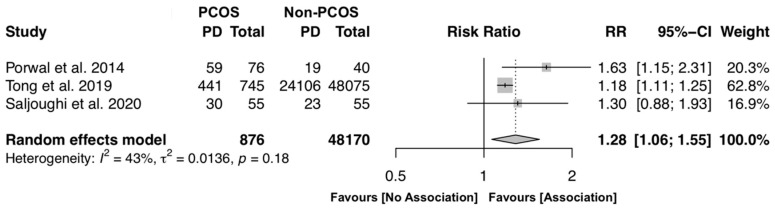
Forest plot of risk ratio of polycystic ovary syndrome (PCOS) female individuals to have periodontitis (PD) (p-value < 0.0001). Mean effect size estimates have been calculated with 95% confidence intervals and are shown in the figure. The size of the squares is proportionate to the study sample size, continuous horizontal lines and diamonds width represents 95% confidence interval. Diamond and the vertical dotted line represent the overall pooled estimate.

**Figure 3 jcm-09-01961-f003:**
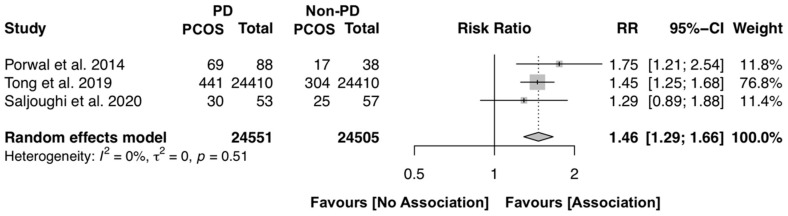
Forest plot of risk ratio of PD female individuals to have PCOS (p-value < 0.0001). Mean effect size estimates have been calculated with 95% confidence intervals and are shown in the figure. The size of the squares is proportionate to the study sample size, continuous horizontal lines and diamonds width represents 95% confidence interval. Diamond and the vertical dotted line represent the overall pooled estimate.

**Figure 4 jcm-09-01961-f004:**
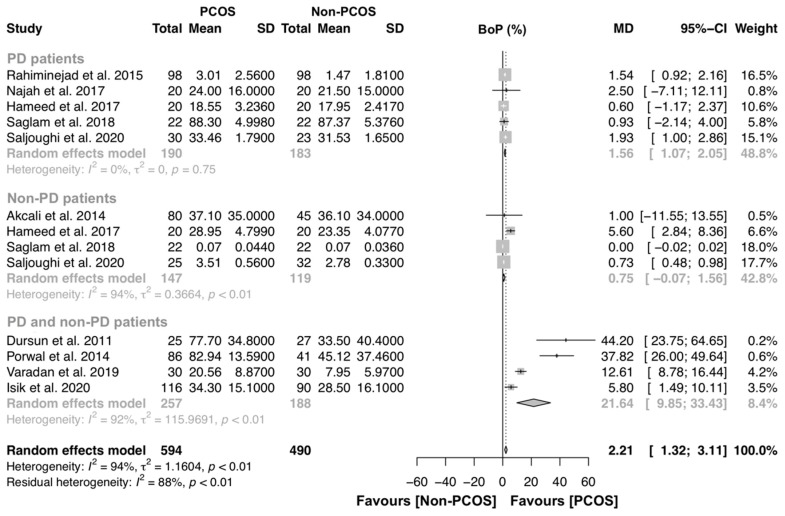
Forest plot of studies evaluating bleeding on probing (BoP) levels in non-PD patients with and without PCOS (p-value < 0.0001). Mean effect size estimates have been calculated with 95% confidence intervals and are shown in the figure. Area of squares represents sample size, continuous horizontal lines and diamonds width represents 95% confidence interval. Diamond and the vertical dotted line represent the overall pooled estimate.

**Figure 5 jcm-09-01961-f005:**
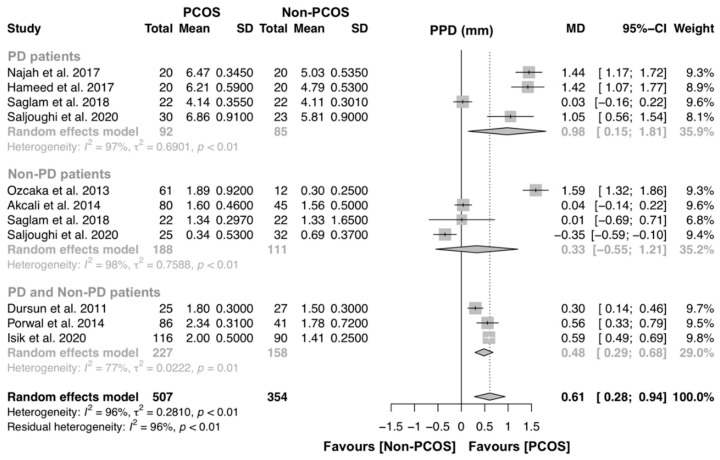
Forest plot of studies evaluating periodontal probing depth (PPD) levels in PD patients with and without PCOS (p-value = 0.0003). Mean effect size estimates have been calculated with 95% confidence intervals and are shown in the figure. Area of squares represents sample size, continuous horizontal lines and diamonds width represents 95% confidence interval. Diamond and the vertical dotted line represent the overall pooled estimate.

**Figure 6 jcm-09-01961-f006:**
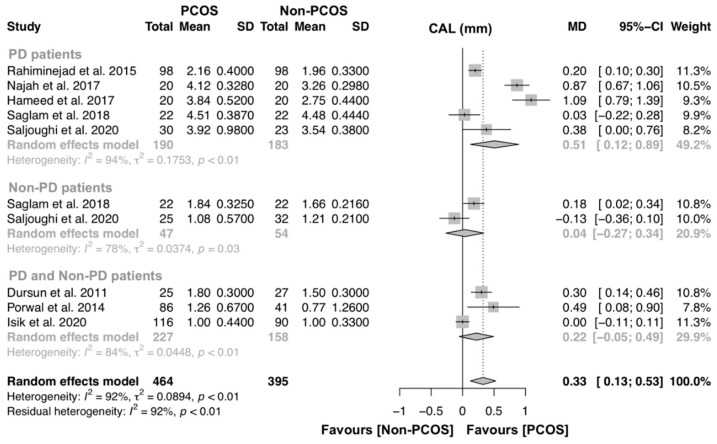
Forest plot of studies evaluating clinical attachment loss (CAL) levels in PD patients with and without PCOS (p-value = 0.0014). Mean effect size estimates have been calculated with 95% confidence intervals and are shown in the figure. Area of squares represents sample size, continuous horizontal lines and diamonds width represents 95% confidence interval. Diamond and the vertical dotted line represent the overall pooled estimate.

**Table 1 jcm-09-01961-t001:** Characteristics of the included studies.

Authors (Year)	Country	Funding	Number of Subjects	PCOS	PCOS–PD	PD	H	Mean Age ± SD (PCOS/PCOS–PD/PD/H)	Exclusion Criteria	PD Diagnostic Criteria	PCOS Criteria
Porwal et al. (2014)	India	None	126	85	0	0	41	23.50 ± 2.67/NA/NA/22.9 ± 4.7	Thyroid dysfunction, hyperprolactinemia, and androgen-secreting tumors to avoid misdiagnosis of PCOS; chronic inflammatory disease; smoking and alcohol habits; Systemic ATB within 3 months; periodontal treatment within 6 months; and AgP diagnosis	Page and Eke 2012	Rotterdam Criteria
Akcali et al. 2014	Turkey	IADR	125	80	0	0	45	25.6 ± 5.2/NA/NA/26.1 ± 4.7	Hyperandrogenism, DM, hyperprolactemia, congenital adrenal hyperplasia, thyroid disorders, Cushing syndrome, HTA, hepatic or renal dysfunction; BMI > 30 kg/m2; CVD; medications (e.g., oral contraceptive agents, steroid hormones, insulin-sensitizing drugs and ATB or ant-inflammatory); periodontal status within the last 6 months	Armitage 1999	Rotterdam Criteria
Hameed et al. 2017	India	None	80	20	20	20	20	NA	Smoking habits; pregnant women; periodontal therapy at the previous 3 months; anti-inflammatory or ATB therapy during the last 3 months; contraceptives or hormonal drugs or medications for PCOS; systemic diseases (e.g., DM, HTA, CVD) which could affect periodontal health	AAP 1999	Rotterdam Criteria
Saglam et al. 2017	India	None	88	22	22	22	22	27.6 ± 4.0/28.6 ± 4.5/28.2 ± 4.3/27.8 ± 3.9	Cushing syndrome, non-classic congenital adrenal hyperplasia, hyperprolactinemia, thyroid dysfunction, and androgen-secreting tumors	Page and Eke 2012	Rotterdam Criteria
Tong et al. 2019	Taiwan	None	48820	304	441	23969	24106	NA	Endocrine disorders (e.g., Cushing syndrome, non-classic congenital adrenal hyperplasia, hyperprolactinemia, thyroid dysfunction and androgen-secreting tumors)	ICD-9-CM code: 523.4	ICD-9-CM code: 256.4X
Saljoughi et al. 2020	Iran	Arak University of Medical Sciences	110	25	30	23	32	45.3 ± 3.0/45.2 ± 3.2/45.3 ± 3.1/45.5 ± 3.3	Interfering drugs (e.g., ATB, oral contraceptives, antihypertensive, and DM drugs); infection in the last 6 months; systemic diseases (e.g., thyroid disorders, hyperprolactinemia, DM, HTA, malignancies, osteoporosis); obesity and overweight; smoking and alcohol habits, and pregnant women	Armitage 1999	Rotterdam Criteria

AgP—aggressive periodontitis; ATB—antibiotics; CVD—Diagnosis of cardiovascular diseases; DM—diabetes mellitus; H—periodontal healthy patients; HTA—hypertension; PCOS—polycystic ovarian syndrome; PD—patients with periodontitis; PCOS-PD—Patients diagnosed with polycystic ovarian syndrome and periodontitis; SD—Standard devidation.

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
