# Peer review of "Is There a Bidirectional Association between Polycystic Ovarian Syndrome and Periodontitis? A Systematic Review and Meta-analysis"

_jcm, 2020, doi:10.3390/jcm9061961_

Round 1

Reviewer 1 Report

Title:

The authors identify the publication as a systematic review and meta-analysis.

Abstract:

It is well structured and includes the key aspects of the work.

Introduction:

It offers a sufficient context to follow the work and includes justification and objectives.

Materials and methods:

It is rigorous and the explanation is sufficient to reproduce the study, it contains all the necessary key aspects. The PECO questions are well constructed and allow to achieve the proposed objectives. Both the search strategy and the article selection process, and the extraction of variables are well explained, as well as the method for evaluating bias risk. Statistical analysis is correct.

Results

The results include the selection of the studies and the flow chart, describe the main findings of the study with confidence intervals and illustrate them with forest plots, and the bias risks of each article are also presented.

Discussion

It is well structured and covers all aspects raised in the work. She puts the argument well to reach the conclusions and explains both the limitations and the implications of the study. The conclusion is adequate according to the results obtained.

Reviewer 2 Report

This is a well conducted meta-analysis and a useful contribution to the literature. The authors recognise the limitations of the available evidence and the paucity of studies available. One possible confounder is whether treatment of the PCOS might influence the presence of PD. For example, metformin, by reducing insulin resistance, might lower the prevalence of PD in these patients. Metformin is widely used in the management of PCOS. Future research should investigate whether particular PCOS treatments influence PD.

page 2 Line 51 "invade our organism" - remove this expression;

suggest change sentence "to allow bacteria and bacterial products to gain access to the systemic circulation through the ulcerated epithelium and destruction of the periodontium"
